# Current and Future Perspectives in the Diagnosis and Management of *Helicobacter pylori* Infection

**DOI:** 10.3390/jcm11175086

**Published:** 2022-08-30

**Authors:** Malek Shatila, Anusha Shirwaikar Thomas

**Affiliations:** Department of Gastroenterology, Hepatology and Nutrition, The University of Texas MD Anderson Cancer Center, Houston, TX 77030, USA

**Keywords:** *Helicobacter pylori*, gastric cancer, peptic ulcer disease, triple therapy, bismuth, vonoprazan, dyspepsia

## Abstract

*Helicobacter pylori* (*Hp*) is a prevalent organism infecting almost half the global population. It is a significant concern, given its associated risk of gastric cancer, which is the third leading cause of cancer death globally. Infection can be asymptomatic or present with dyspeptic symptoms. It may also present with alarm symptoms in the case of progression to cancer. Diagnosis can be achieved non-invasively (breath tests, stool studies, or serology) or invasively (rapid urease test, biopsy, or culture). Treatment involves acid suppression and regimens containing several antibiotics and is guided by resistance rates. Eradication is essential, as it lowers the risk of complications and progression to cancer. Follow-up after eradication is similarly important, as the risk of cancer progression remains. There have been many recent advances in both diagnosis and treatment of *Hp*. In particular, biosensors may be effective diagnostic tools, and nanotechnology, vaccines, and potassium-competitive acid blockers may prove effective in enhancing eradication rates.

## 1. Introduction

*Hp* is a Gram-negative, characteristically curved bacteria that was first observed to be present in the stomach lining in the 19th century [1,2]. It was not until 40 years ago that its association with gastric inflammation was demonstrated [3]. The finding that *Hp* could cause gastritis earned the investigators the Nobel Prize in 2005 for its myriad implications [2]. This discovery generated a massive body of research into the manifestations, treatments, and associations of *Hp* infection and revolutionized our understanding of the role of pathogens in disease [2,4].

### Why Is H. Pylori Important?

*Hp* is strongly associated with duodenal ulcers (present in as many as 90% of cases), gastric ulcers (up to 80%), and malignancy; it can lead to mucosa-associated lymphoid tissue (MALT) lymphoma, as well as gastric cancer in as many as 90% of cases [5]. In 2014, the World Health Organization (WHO) called for the elimination of *Hp* as a means to decrease gastric cancer mortality worldwide, and in 2017. it deemed clarithromycin-resistant *Hp* strains a serious threat to public health [6]. In this review, we will present an overview of *H. pylori* disease characteristics, including epidemiology and clinical presentation, and discuss the most recent advances in evaluation and management of this entity.

## 2. Epidemiology

*Helicobacter pylori* is highly ubiquitous [7], colonizing roughly half of the world’s population [7,8]. The primary mechanism of transmission is yet to be identified but is presumed to involve person-to-person transmission [9] through fecal/oral exposure. The prevalence of *Hp* varies widely by region; Asia, Latin America, and Africa tend to have higher rates (up to 80% in some countries), whereas North America and Oceania have the lowest rates (as low as 24%) [10,11]; those born in the 1930s have a much higher prevalence than those born in the 1970s [10]. One meta-analysis estimated the global prevalence of *Hp* at 44.3%, ranging from 34.7% in developed countries to 50.8% in developing countries [12], and most recent studies show a continuous decline in *Hp* prevalence over the years [13]. Despite this trend, there is a worrisome increase in antibiotic-resistant *Hp* strains [14].

### 2.1. Antibiotic Resistance

Unsuccessful eradication has been a recurrent issue over the years; treatment failure rates are continuously on the rise, in large part due to the increasing prevalence of antibiotic-resistant *Hp* strains [14,15,16,17,18]. This is a global phenomenon affecting most countries, although specific resistance rates vary by region and antibiotic type [19]. One meta-analysis found that clarithromycin resistance reached as high as 35% in eastern Mediterranean, European, and western Pacific regions, whereas it was the lowest in Africa, the Americas, and southeast Asia, at around 15% [14]. Levofloxacin showed a somewhat similar trend, with 14% resistance in the Americas, Africa, and Europe; and around 25% in Mediterranean, southeast Asian, and western Pacific regions. Reported metronidazole rates were much higher in this study, ranging from 30 to 91%, whereas amoxicillin resistance was negligible in most regions, except for Africa, where it was 38% [14]. To put these numbers into perspective, a local resistance of >15% is the common threshold for choosing alternate treatment regimens [19,20]. In reality, the epidemiology of antibiotic resistance is far more complex, as there considerable variation within the countries of each region. This obstacle is compounded by the fact that in most regions worldwide, studies are focused primarily on a handful of countries [19]. In the US, for instance, national data are scarce, and fewer than half the states are routinely included in studies [21,22]. Better data are available from Europe, where various studies have been conducted in individual countries, as well as larger-scale projects. One study on 3974 patients by the European Registry of *Helicobacter pylori* Management (*Hp*-EuReg) found that resistance rates to clarithromycin and levofloxacin were significantly higher in southern Europe (e.g., Italy, Spain, and Greece) as opposed to northern Europe (e.g., Norway) [23]. Alarmingly, strains collected from 52% of *Hp*-naïve and 80% of non-naïve patients exhibited some form of antibiotic resistance in this study [23]. Antibiotic resistance may, in part, be influenced by treatment for previous *Hp* infections. This entity recurs in around 4% of cases annually [24,25], and various risk factors have been identified that contribute to this phenomenon.

### 2.2. Risk Factors

Risk factors can be categorized on a societal or individual level. The former encompasses geographic location; economic development; and sanitation, including access to clean food and water [26]. Low familial socioeconomic status and overcrowding (i.e., crowded living conditions and large family sizes) are also associated with increased *Hp* prevalence [26]. Consumption of unpasteurized dairy products [27], sheepherding [28], high-risk occupations (healthcare) [29], obesity [30], male gender [31], and the gut microbiome [32] pose an increased risk of infection. Smoking and alcohol are two variables that are controversial with respect to their role in *Hp* infections [33,34,35,36,37,38,39,40,41,42,43].

## 3. Etiopathogenesis

The pathogenesis of *Hp* infection can be divided into distinct steps, whereby the bacteria (1) attaches to and colonizes the gastric mucosa, (2) evokes and evades an immune response, and (3) induces disease. Once present in the stomach, *H. pylori* swims toward the mucous lining the epithelial layer, showing a tropism for sites of injury along the stomach wall [44]. This chemotaxis relies on Tlp receptors (mainly TlpB) to direct flagellar motion in response to chemical signals in the cell environment [45]. Urea, gastric acid, lactate, and reactive oxygen species have all been identified as signals for these receptors; urea in particular is secreted by the gastric epithelium and is thought to play a significant role in bacterial colonization [45]. However, undiscovered chemicals may also be implicated in this process [46]. *Hp* utilizes urease to protect itself from the surrounding acid environment. Urease breaks urea down into ammonia and other useful metabolites, increasing the pH in the microenvironment to create a thin, pH-neutral layer around the bacterial cell, allowing it to survive the gastric acid. This barrier reduces the viscosity of the mucin gel lining the stomach wall and allows the bacteria to move freely through the mucous toward the gastric glands that it will ultimately colonize [46,47].

Bacterial attachment to gastric epithelial cells is a complex process that involves the synergistic interaction of several elements and relies heavily on Lewis (Le) antigens. Lewis antigens are cell-surface glycoproteins that mediate cell-to-cell adhesion by binding to selectins on target cells [48]. The lipopolysaccharide (LPS) component of the *Hp* cell wall has been found to express Lewis-like antigens, among which Le^X^, in particular, has been shown to play a minor role in adhesion [49].

Bacterial outer membrane proteins (OMPs), on the other hand, operate as binding sites to which host Lewis antigens can bind, facilitating attachment. The OMPs on *Hp* can be divided into five genomic families [50], of which *H. pylori* OMP (Hop) and *Helicobacter* outer membrane (Hom) play the largest roles [51]. Blood antigen-binding adhesin A (BabA) and sialic acid-binding adherence (SabA) are members of the Hop family and are the most well-studied of all OMPs [52]. BabA promotes cell-to-cell adhesion by binding to host Le^B^ [53], whereas SabA binds to sialylated Le^X^ (sLe^X^) to facilitate cell adhesion [54]. SabA additionally stimulates a neutrophil response by binding neutrophil sLe^X^ and activating a G-protein-coupled, receptor-mediated signaling cascade [55]. Interestingly, sLe^X^ is upregulated upon the occurrence of *Hp* infection and gastric inflammation, which suggests that SabA may be involved in strengthening and maintaining adhesion, as opposed to initiating it [54]. MUC5a and MUC1 mucin receptors are the primary target for these OMPs and can expedite and hinder infection, respectively [56,57,58]. Whereas BabA and SabA are the main adhesins involved, several other OMPs, such as outer inflammatory protein A (OipA), HopQ, HopZ, and the Hom family, improve *Hp* adhesion and promote inflammation by prompting the transcription of virulence factors and the secretion of inflammatory cytokines [52].

Certain virulence factors involved in pathogenicity also contribute to adhesion. BabA-Le^B^ binding has been shown to activate the type-four secretion system (T4SS), a pilus-like structure that allows for the translocation of effector proteins, such as cytotoxin-associated gene A (CagA) and vacuolating cytotoxin A (VacA) [54]. CagA binds to epithelial cell integrin-β1 [59] to anchor *Hp* and hijack host signaling pathways to disrupt cell motility, proliferation, and cytoskeletal stability [60]. VacA, on the other hand, binds to a wide range of receptors, with numerous downstream effects. It primarily functions as a pore-forming toxin than acutely induces host cell apoptosis but plays a key role in evading the immune response in chronic infections. It carries out this latter role by impairing autophagy and forming intracellular vacuoles in the host cell in which *H. pylori* can survive. It also binds to the integrin β2 subunit on T cells to inhibit their activation and proliferation and can induce macrophage apoptosis by inhibiting interferon-β signaling [61]. Together, these toxins are the major virulence factors expressed by *Hp* that are key to its pathogenicity. Whereas VacA is expressed by virtually all *Hp* cells, CagA is only present in specific strains; interestingly, CagA positivity is associated with more severe infection and worse clinical outcomes, including an increased risk of future malignancy [60,62]. Although these are the two major proteins involved in the pathogenesis of *Hp* infection, there are myriad others that similarly aid in adhesion, immune evasion, and provocation of inflammation [60].

## 4. Clinical Presentation

The presentation of *H. pylori* infection is highly variable. As many as 90% of individuals carrying the bacteria are asymptomatic [63]. It can present as dyspepsia, defined as an epigastric discomfort or pain lasting longer than one month that may be associated with nausea, early satiety, epigastric fullness, and bloating, among other symptoms [64], which is often presumed to be functional in etiology [65]. Gastrointestinal clinical symptomatology of *Hp* infection poorly correlates with severity of gastric mucosal injury upon endoscopy [19,64]. *Hp* can therefore go unnoticed and untreated, in which case it progresses to chronic gastritis [66]. This chronic inflammation of the gastric epithelium can promote intestinal metaplasia, which predisposes to gastric cancer [64]. Despite being asymptomatic for years, such patients may present with alarm symptoms of weight loss, iron deficiency anemia, dysphagia, vomiting, and the presence of an abdominal mass [19,64,65]. Similarly, MALT lymphomas may develop as a result of gastritis and present with dyspeptic or non-specific constitutional symptoms [64,67]. *Hp* can also cause peptic ulcer disease (PUD), with the risk of complications, such as gastrointestinal bleeding and perforation [64].

Rarely, *H. pylori* infection may present with extragastrointestinal manifestations, such as isolated iron deficiency anemia; idiopathic thrombocytopenic purpura; and ocular, dermatological, and metabolic diseases [65,68].

## 5. Evaluation and Management:

### 5.1. Indications for Testing

The American College of Gastroenterology (ACG) recommends testing in the following cases [65]:(1)All patients with active PUD;(2)All patients with a previous history of PUD (unless there is documentation of a resolved prior *Hp* infection), low-grade MALT lymphoma, or a history of endoscopic resection of early gastric cancer;(3)Patients with uninvestigated dyspepsia under the age of 60;(4)Patients initiating long-term, non-steroidal, anti-inflammatory drugs;(5)Patients with unexplained iron deficiency anemia despite appropriate workup; and(6)Adults with idiopathic thrombocytopenic purpura.

Other expert panels, including the global Taipei consensus [69] (TC) and the Houston conference [70] (HC), support guidelines similar to those outlined above, with the addition of a few other indications:(7)Family members residing in the same household as patients with proven active *Hp* infection (HC);(8)Patients with a family history of PUD or gastric cancer (HC);(9)First-generation immigrants from high-prevalence areas *or* high-risk groups (HC); and(10)Populations with a high incidence of gastric cancer (TC).

In all of the above cases, a “test-and-treat” strategy is recommended to ensure eradication of the bacteria and reduce the severity of symptoms and the risk for carcinogenesis. A variety of tests can be used to diagnose infection with *H. pylori*, and they can be divided into non-invasive and invasive tests, a summary of which can be found in Table 1.

### 5.2. Non-Invasive Tests

The most commonly used test to identify *H. pylori* is the urea breath test (UBT), which measures the difference in proportion of ^13^C/^14^C in exhaled air before and after the patient swallows radioactively labeled urea. This relies on the previously discussed *Hp* urease, which generates radiolabeled ^13^C carbon dioxide as a result. Patients with active *Hp* infection exhale higher quantities of ^13^C than healthy patients. Typically, four respiratory samples are collected (two before and two after ingestion of urea), and the labeled carbon dioxide is detected using mass spectrometry [71]. UBT is an accessible and commonly used tool to diagnose *Hp* infection, with a recent meta-analysis showing sensitivity and specificity of around 95% [72,73]. However, results can be influenced by concurrent medications, so patients are required to stop antibiotics 30 days prior to the test and proton pump inhibitors (PPIs) 15 days prior, as they may produce false negatives [71].

Stool antigen testing (SAT) is another low-cost and accurate means of diagnosing *Hp* infection that is often preferred by patients and physicians for its simplicity [71]. There are two main types of stool tests: enzyme immunoassays (EIAs) and immunochromatography assays (ICAs). Both tests operate under similar principles. In EIAs, a solution containing monoclonal or polyclonal antibodies is added to a diluted stool sample and processed, antigens detected through spectrometry [74]. In immunochromatography, antigen detection is made possible by a reaction caused by the antigen–antibody complex, producing a visible color change in the medium [75]. Between these two testing methods, EIA has proven to be a more effective diagnostic tool, although the accuracy of SATs varies widely depending on the detection kit used [76,77].

The final non-invasive method for diagnosing *H. pylori* is serological testing for immunoglobulin G (IgG) antibodies against *Hp* using an enzyme-linked immunosorbent assay that operates in a similar fashion to SAT. IgG antibodies appear about three weeks after infection and can remain detectable for several years thereafter. Serological tests have fallen out of favor for this reason, as they have trouble distinguishing past infections from recent or active infections and are associated with a risk of false positives [78]. Despite a high reported specificity and sensitivity for serologic antibody tests ranging from 80 to 95%, these values vary considerable depending on the testing kit used, with values reported as low as 55% [72,78]. This is especially true in the case of the newer latex agglutination immunoassay test. The latex or LZ test is increasingly used due to its low cost and rapid processing ability [79]. However, it relies on the agglutination of latex-bound antigens, and interpretation of results is highly subjective, potentially increasing the risk of false positives [79]. To their credit, serological tests are not affected by recent PPI or antibiotic use, making them an option for patients who have used either.

### 5.3. Invasive Tests

Endoscopic assessment is a critical component of invasive testing for *H. pylori* upon which all other invasive tests depend [71]. Conventional endoscopy alone is, for the most part, inadequate in diagnosing *Hp* infection, and a biopsy with histological assessment is still mandatory for evaluation [19]. However, advancements in endoscopic technology have allowed for image-enhanced scoping to improve the accuracy of endoscopic evaluation [19,80]. There is one scoring system currently in use called the Kyoto classification [81] that is used to evaluate active *H. pylori* infection and risk of gastric cancer. It consists of five endoscopic findings (atrophy, intestinal metaplasia, enlarged folds, nodularity, and diffuse redness), cumulating in a score ranging from 0 to 8. A Kyoto score ≥2 indicates *H. pylori* infection, whereas a score ≥4 suggests gastric cancer risk [80]. Whereas a few studies have supported the accuracy of the Kyoto classification in diagnosing active *H. pylori* infection [81,82], endoscopy is rarely the only diagnostic technique used to evaluate *Hp* infection. Instead, it is almost always paired with biopsies or alternative tests.

The gold standard for *H. pylori* diagnosis is histological examination. For this evaluation, at least six biopsies must be taken during biopsy targeting the antrum, large and small curvatures of the stomach, and the middle of the gastric body, as well as any suspicious lesions or ulcerations [71]. Hematoxylin–eosin and Giemsa stains are the most inexpensive and commonly used stains, but immunohistochemical staining is the most accurate (fluorescence in situ hybridization) and is recommended when histochemical methods fail [83]. The updated Sydney grading system relies on histopathological findings to assess the severity of chronic gastritis and categorizes the intensity of mononuclear inflammatory cellular infiltrates, polymorph activity, atrophy, intestinal metaplasia, and *Hp* density as mild, moderate, or severe [84]. The sensitivity and specificity of histological methods can range from 60% to 100% and depend on a variety of factors, including stain used; location, size, and quality of the sample; and the pathologist’s experience [71].

Alternative testing methods include bacterial culture, molecular testing (polymerase chain reaction; PCR), and rapid urease test (RUT). Bacterial culture is a highly specific means of diagnosing *H. pylori* infection but, as mentioned earlier, can be an arduous task that requires well-equipped laboratories [71]. *Hp* is a notoriously difficult micro-organism to grow and requires incubation for more than a week in selective blood agar (a detailed description of culture requirements can be found in the study by Blanchard and colleagues) [85]. It has the additional benefit of identifying antibiotic resistance, as does PCR testing, which is becoming increasingly necessary [71]. The RUT, on the other hand, relies on a pH indicator that changes color in response to the ammonia produced by *Hp* urease. Previous studies have shown it to be highly sensitive and specific, at ≥90%, and it has the added benefit of rapidly producing results (within 5 min with some tests). Given its ease of use, it is considered the first-line diagnostic method in cases for which endoscopy is indicated [71].

**Table 1 jcm-11-05086-t001:** Sensitivities and specificities of various diagnostic modalities.

Test	Sensitivity	Specificity	Cost [86]	Advantages [87]	Disadvantages [87]	Study
UBT	97%	96%	Cost-effective	Fast, simple, non-invasive, good for detecting eradication	Potential risk for false negatives in cases of bleeding and PPI or antibiotic use; low accuracy in atrophic gastritis and gastric malignancy	Abd Rahim et al., 2019 [72] Zhou et al., 2017 [88]
Fecal antigen test	94%	97%	Cost-effective	Fast, simple, inexpensive, can potentially be used to determine antibiotic sensitivity	False negatives in cases of low bacterial load; accuracy affected by recent antibiotic, bismuth, or PPI use; may be uncomfortable for patients; difficulty maintaining sample; and variable accuracy depending on commercial kit used	Gisbert et al., 2006 [89]
Serology	Variable (76–84%)	Variable (79–90%)	Cost-effective	Cheapest, widely available, can be used in patients with recent PPI or antibiotic use	Failure to distinguish between acute and previous infection; cannot confirm eradication	Thaker et al., 2016 [90]
Rapid Urease Test	Variable (80–99%)	Variable (92–100%)	Cost-effective	Fast, inexpensive, simple	Accuracy impaired by gastric ulcer bleeding or intestinal metaplasia; invasive	Roy et al., 2016 [91]
Bacterial Culture	Variable (70–80%)	100%	Expensive	Determination of antibiotic resistance and sensitivity	Expensive, time-consuming, requires a well-equipped lab	Thaker et al., 2016 [90]
PCR	96%	98%	Expensive	High sensitivity and specificity; effective, even at low bacterial loads	Expensive, requires a well-equipped lab, false-positive risk due to detection of DNA from dead bacteria	Pichon et al., 2020 [92]

### 5.4. Management

Most societies endorse the non-invasive “test-and-treat” method [93,94,95], and initial endoscopy is recommended for older patients and those who present with alarm symptoms (first-degree relative with upper GI malignancy, weight loss, GI bleeding, dysphagia, odynophagia, persistent vomiting, and abnormal imaging) [94,95].

The first-line management of confirmed *H. pylori* infections utilizes a PPI, alongside 2–3 antibiotics for periods ranging from 3 to 14 days [64]. A full breakdown of the available first- and second-line treatment modalities can be found in Table 2.

Local antibiotic resistance rates are a key factor in determining the most appropriate first-line of management, as reflected by several major guidelines [20,64,65,96,97]. Furthermore, reuse of antibiotics from first-line treatments in subsequent therapy can lead to secondary antibiotic resistance and should be avoided [98]. Resistance to nitroimidazole antibiotics, such as metronidazole, are interesting in that whereas resistance rates can be high, this resistance can be overcome with dose adjustments and the addition of bismuth, allowing for its reuse in new regimens after an initial failure [98]. Whereas guidelines are not clear on sensitivity testing for first-line management of *Hp*, susceptibility testing is recommended in refractory cases [98].

An important consideration in initial management of infection is that the most commonly used acid suppressants, PPIs, are metabolized by hepatic cytochrome P450 (CYP2C19) [99]. Genetic polymorphisms may limit the efficacy of treatment regimens with particular PPIs [100]. It may be advisable to rely on PPIs less affected by CYP metabolism, such as esomeprazole and rabeprazole, especially in non-Asian regions, where extensive metabolizers are common [66,98].

The exact recommendation for treatment duration differs depending on the guidelines and treatment line (Table 2). However, there is a trend toward standardization of all treatment lines to 14 days [108].

Table 2 highlights various treatment strategies: sequential, hybrid, concomitant, and reverse hybrid. Sequential therapy involves initial dual treatment with a PPI and amoxicillin for 5–7 days, followed by standard triple therapy for the same amount of time [108]. Concomitant therapy is a non-bismuth quadruple therapy and involves the concurrent administration of four medications [109]. Hybrid therapy combines the two, initiating patients on dual therapy (PPI + amoxicillin) for 7 days, then adding two more antibiotics for the next 7 days. Reverse hybrid therapy follows the opposite order: three antibiotics and a PPI for 7 days, followed by only a PPI + amoxicillin [110]. Sequential therapy has fallen out of favor due to worse eradication rates [20,64,66,95,96,108,109]. Hybrid and reverse-hybrid treatment strategies have proven to be equivalent to concomitant across various studies but are of limited efficacy in areas of high dual resistance to clarithromycin and metronidazole [111,112]. Concomitant therapy is currently the most widely used treatment strategy.

### 5.5. Treatment Outcomes

The ultimate goal of treatment is documented eradication of the bacteria [20,65,96,97]. There is evidence for endoscopic and histologic remission of gastritis features, including intestinal metaplasia and reduction in recurrence following eradication therapy [69,113,114,115,116,117,118,119,120], as well as prevention of gastric adenocarcinoma and regression of gastric MALT lymphoma [121,122,123,124,125,126,127]. Treatment also resolves *H. pylori*-associated iron deficiency anemia [128] and ITP [129].

## 6. Long-Term Surveillance and Complications

### 6.1. Surveillance

It is currently recommended to retest at least 4 weeks after completion of the initial treatment regimen, with the patient stopping PPIs as many as 2 weeks prior [64]. All aforementioned diagnostic tests are suitable for confirmation of eradication.

Despite the lack of established evidence-based guidelines, a growing body of literature supports endoscopic surveillance following eradication, particularly in high-risk patients [130,131,132,133]. The “ABC method” relies on an investigation of anti-*H. pylori* antibodies and serum pepsinogen (PG), whereby patients are divided into four groups depending on the presence of either (group A, negative for both; group B, anti-*Hp*-positive and PG-negative; group C, positive for both; group D, anti-*Hp*-negative and PG-positive). Groups B, C, and D were found to be increasingly more likely to develop gastric cancer than group A and were therefore recommended triennial, biennial, and annual endoscopic follow-up based on the increased risk [134]. Nonetheless, regular endoscopy is invasive, costly, and impractical in certain settings. A recent study identified several biomarkers that could potentially be used in lieu of endoscopy for screening, detection, and monitoring of individuals at risk of gastric cancer (GC), namely virulence markers, genomic markers, transcriptomic markers, and inflammatory markers [135]. cagA and VacA-toxin expression; pepsinogen levels (PG1 and PG2); bacterial lipopolysaccharides; connexin expression; specific microRNA fragments; and cytokines, such as IL-1β, IL-6, IFN-γ, and IL-10, have been found to be significantly upregulated in cases of GC [136,137,138,139].

### 6.2. Complications

Treatment of *H. pylori* does not guarantee permanent eradication. One prospective study of 1050 patients estimated recurrence rates at one and three years to be 1.75% and 4.61%, respectively [140]. Recurrent infection necessitates alternative antibiotic treatment regimens to those used previously and may contribute to increased antibiotic resistance globally. Oral colonization of *H. pylori* is a potential source of reinfection that can often go undiagnosed by standard diagnostic methods for *Hp* and is unaffected by traditional treatment methods, requiring specific treatment strategies [141,142]. The existence of a secondary *H. pylori* colonization site is still highly controversial but may be a consideration in complex cases with frequent recurrences [141,142]. The main complications of untreated *H. pylori* infection were outlined above but include gastric ulceration; perforation; progression to gastric malignancy (adenocarcinoma or MALT lymphoma); and extragastrointestinal manifestations, such as ITP and iron deficiency anemia [65]. Other complications take the form of side effects to treatment and are listed in Table 3.

## 7. Treatment Challenges

*H. pylori* eradication rates may not be optimally attributed primarily to antibiotic resistance and patient noncompliance/adherence due to side effects and provider prescriptive error [146,147,148]. These data highlight the importance of improved awareness and strict adherence to existing guidelines.

## 8. Recent Advances

In the decades since its discovery, considerable progress has been made with regards to diagnostic and therapeutic modalities in the management of *Hp* infection. Nanotechnology is an exciting innovation in the realm of diagnosis and treatment that could eventually represent cost-efficient and less invasive alternative to current endoscopic measures. Biosensors are one such tool that can translate unique biological elements attached to a transducer surface into detectable signals [149]. This is accomplished without additional reagents, reactions, or sample pretreatment, contrary to other modalities, such as PCR or ELISA, while providing accurate and real-time monitoring of disease [149,150,151]. Whereas the process itself is complex, it relies on detecting bacterial antigens or patient *Hp* antibodies. Electrochemical sensors rely on a change in electric potential or conductance of a transducing surface upon element attachment, optical sensors rely on a change in fluorescence or color absorbance, piezoelectric sensors rely on a change in acoustics, and thermal sensors rely on a change in temperature to detect disease [151]. Yadav et al. (2022) mentioned the novel use of aptamers, single stranded nucleic acid sequences that are highly specific to target antigens, proteins, or antibodies, to diagnose disease and have high hopes for their clinical utility [152].

Recent advances in therapy make potential use of nanotechnology as auxiliary treatment, improving drug delivery, with a direct antibacterial effect [153]. The development of novel drugs can also improve treatment of *Hp*. A potassium-competitive acid blocker, vonoprazan (VPZ) has advantages over traditional acid suppression with PPIs, in that it does not require activation by gastric acid and has a longer half-life than PPIs [154]. It is also unaffected by genetic polymorphisms in CYP450 [99]. Genetic polymorphisms are receiving increasing attention for their potential role in treatment outcomes. A few studies to date have shown that eradication rates may be higher in slower CYP metabolizers [99,155,156], although this association has not always been statistically significant. Polymorphisms in immune response genes can similarly impact disease severity and predispose to complications [157,158,159]. VPZ was only recently introduced in east Asia, and several meta-analyses show the superior efficacy of VPZ relative to standard PPI-containing triple therapy [154,160,161,162]. A more recent meta-analysis of RCTs found that VPZ demonstrated comparable and even superior eradication rates relative to PPI across different treatment regimens and in both low- and high-clarithromycin-resistance areas [162]. A lower rate of adverse events was also reported among VPZ users [162].

Antimicrobial peptides are short, positively-charged peptide chains that disrupt the integrity of the negatively charged bacterial cell membrane, leading to cell lysis and disruption of intracellular processes [163]. Photodynamic therapy relies on microbial production of photosensitive molecules that use light to produce cytotoxic reactive oxygen species, leading to bacterial cell death [164]. Phage therapy uses bacteriophages specific to *H. pylori* to induce bacterial cell lysis, eliminating the pathogen [165]. Finally, vaccination is an attractive strategy for combatting *Hp* infection globally. Various attempts have been made to develop an *Hp* vaccine over the years, but the results have been disappointing [166]. Whereas some vaccine candidates have shown potential as an option for prophylaxis, none have yet shown a therapeutic effect [166,167].

## 9. Conclusions

*H. pylori* is a ubiquitous and complex organism that has rightfully received tremendous interest over the years. It can manifest in a variety of ways and increases the risk of severe complications, such as peptic ulceration and malignancy. Therefore, treatment with adequate follow-up is imperative. In the current era of antibiotic stewardship, it is important to be mindful of antibiotic resistance and susceptibility when selecting a treatment regimen. Extensive research has been conducted on the pathogenesis of *Hp* infection, which has aided in identifying diagnostic and therapeutic targets. However, there is still room to improve our knowledge. In particular, there is more to be gleaned regarding bacterial transmission, reinfection, and optimized surveillance. Finally, there have been numerous recent technological advances that hold promise for streamlining the management of this pathogen in the future.

## Figures and Tables

**Table 2 jcm-11-05086-t002:** Treatment options for *Hp*.

First-Line Treatments					
Regimen	Dosing Frequency	Duration	Indications	Notes	Study
Triple PPI (variable dose) ^a^ CA (500 mg) AM ^b^ (1 g) OR MZ (500 mg)	BID BID BID	14 days	First-line treatment in regions where CA resistance is low (<15%) or with high proven local eradication rates (>85%) and in patients with no previous macrolide exposure.	A few studies are listed to summarize global resistance rates [14,101]. Eradication rates of up to 92.6% have been reported with triple therapy when potent and long-lasting gastric acid inhibitors, such as K^+^-competetive blocker vonoprazan, are used [102].	Maastricht ACG Toronto
Concomitant PPI (variable dose) ^a^ CA (500 mg) AM (1 g) MZ (500 mg)	BID BID BID BID	10–14 days	First-line treatment, especially in regions where CA resistance is high (>15%) and metronidazole resistance is low.		Maastricht ACG Toronto
Quadruple Bismuth PPI (variable dose) ^a^ Bismuth (variable dose and preparation) ^c^ AM MZ	QID BID TID or QID	10–14 days	First-line treatment, especially in regions where CA and MZ resistances are high.		Maastricht ACG Toronto Chinese Spanish
**Second-line treatments**					
Quadruple Bismuth PPI (high dose) ^a^ Bismuth (variable dose and preparation) ^c^ TZ (500 mg) MZ (500 mg)	BID QID QID TID to QID	10–14 days	Can be used as a first-line treatment. Used as a second-line treatment if: Triple or concomitant treatment failed; or Earlier bismuth quadruple treatment failed (two different antibiotics need to be used).	The list of antibiotics that can be used alongside bismuth includes [103]: Clarithromycin; Amoxicillin; Doxycycline; Nitroimidazole; Furazolidone; and Levofloxacin.	Maastricht ACG Toronto Chinese Spanish AGA
Levofloxacin Regimens Levofloxacin (500 mg) Amoxicillin (1 g) PPI (high dose) ^a^	QD BID BID	10–14 days	Potential first-line treatment in areas with low fluroquinolone resistance (reference to ACG). Second-line treatment after failure of a bismuth regimen.	Only ACG suggests this as a first-line treatment in regions where levofloxacin resistance is low. Levofloxacin can be replaced by sitofloxacin.	Maastricht ACG Toronto Chinese Spanish AGA
**Rescue treatments ^d^**					
High-dose dual PPI (high dose) ^a^ AM (750 mg or 1 g)	BID QID or TID respectively	14 days	Salvage therapy after two eradication failures.	Amoxicillin resistance rates are still low globally.	ACG Toronto AGA
Rifabutin-based triple Rifabutin Amoxicillin PPI (high dose) ^a^	QD TID BID	14 days	Salvage therapy after two (Malfertheimer) or three (fallone) eradication failures. AGA guidelines suggest use as a second-line treatment after failed Bismuth therapy.	There is some concern about increasing M. tuberculosis resistance as a result of this treatment.	Maastricht Toronto AGA Spanish
**Alternative or adjunctive treatments**					
Statins Atorvastatin (40 mg) Simvastatin (20 mg)	QD BID	14 days	Experimental use	Statins have been shown to have antibacterial and anti-inflammatory effects [104,105]. One study found that statins reduced *Hp* burden in macrophages and increased *Hp*-infected macrophage autophagy [106].	AGA
Probiotics ^e^		14 days	Experimental use	Probiotic strains have been shown to have a beneficial effect on eradication and to reduce treatment adverse effects, including: *Lactobacillus*; *Bifidobacterium*; *Lactiplantibacillus*; and *Saccharomyces*	AGA Kyoto Viazis et al., 2022 [107]

Notes: Abbreviations: PPI—proton pump inhibitor; CA—clarithromycin; AM—amoxicillin; MZ—metronidazole; BID—bidaily; QID—quad daily; TID—tridaily; TZ—tetracycline; QD—once daily; ^a^ dose varies depending on PPI used. Standard doses include esomeprazole 20 mg, lansoprazole 30 mg, omeprazole 20 mg, pantoprazole 40 mg, and rabeprazole 20 mg. High dose implies double the standard dose. ^b^ In patients with a penicillin allergy, amoxicillin should be substituted for metronidazole. ^c^ Bismuth can come in multiple preparations; the most common preparations are: Bismuth subsalicylate (262 mg), two tablets QID; colloidal bismuth subcitrate (120 mg), one tablet QID; bismuth biskalcitrate (140 mg), three tablets QID; bismuth subcitrate potassium (140 mg), three tablets QID. ^d^ Alternative antibiotics that can be used in rescue treatments include sitafloxacin, tinidazole, and furazolidone. ^e^ Lactobacillus and Bifidobacterium are supported by a growing body of evidence, whereas the benefits of Lactiplantibacillus and Saccharomyces are supported by a limited number of studies.

**Table 3 jcm-11-05086-t003:** Medication adverse events resulting from *H. pylori* treatment.

Adverse Effect	Reported Frequency *	Associated Treatment Group ^†^
Taste disturbance/oral mucositis [143,144,145]	17–44%	Triple therapy
Nausea [143,144]	7–31%	Bismuth
Diarrhea [143,144,145]	7–33%	Triple
Dyspepsia [143,145]	3–11%	
Reduced appetite [143,144]	4–12%	
Vomiting [144,145]	3–6%	
Abdominal pain [144,145]	8–20%	
Headache [144,145]	7–31%	Bismuth
Rash [144]	3–7%	Bismuth
Discoloration of feces [144,145]	4–16%	Bismuth
Oral/vaginal candidiasis [143,145]	1–4%	

Notes: * frequencies represent a range report in the selected studies comparing different treatment modalities; † not based on statistical comparison between different treatments, rather an observed difference in frequency between different treatment modalities. However, Calvet et al. [143] showed that triple therapy was significantly associated with more frequent taste disturbance.

## Data Availability

Not applicable.

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
