# Peer review of "Current and Future Perspectives in the Diagnosis and Management of Helicobacter pylori Infection"

_jcm, 2022, doi:10.3390/jcm11175086_

Round 1
Reviewer 1 Report
The present review is focused on the current and future perspectives in the diagnosis and treatment of Helicobacter pylori infection.
The manuscript includes a good revision of this topic, highlighting some very interesting points.
My first comment is regarding the terminology used by the authors. Being a species name, Helicobacter pylori should always be written in italic. I want to ask the authors to correct this through all the manuscript. As an abbrevation, H. pylori or Hp are usally used. Please apply this to the manuscript.
The section "History of Helicobacter pylori" is a little bit too long, and somehow a little "childish". I believe that there is no need of such detailed description of discovery of the bacteria.
The section "Etiopathogenesis" need to include more information regarding the colonization of the human gastric mucosa. Not only BabA is involved in the binding to the gastric cells. A lot of other OMPs (outer membrane proteins) are involved, being this a two-step process. There are several review articles describing very well this process and the role of glycans for a sucessful colonization. Please include a more careful description of the infection mechanism.
In general, the manuscript provides a review of the current diagnostic techniques used for H. pylori detection. However, since this one of the main focus of the manuscript, I would like to have a more detailed description of each technique on the sub-sections "non-invasive and invasive tests". Besides this, I believe that would be more appealling if a description of the advantages and disadvantages of each technique is included on Table 1. This would make much easier to compare the different techiques.
The sub-section "How to treat" should be removed since this is just repeated information.
The Table 3 seems to appear from nowhere in the manuscript. There is no conection to any part of the text (is not mentioned in the text). Please include an introductory text and link this table to text.
The manuscript provides a good statement of the future perspectives regarding Hp treatment.
Reviewer 2 Report
- Helicobacter pylori should be specified with italic font throughout the manuscript.
- the author should give more details about "a study on 3,974 patients by the European Registry of Helicobacter pylori Management (HP-EuReg) found that strains collected from 52% of HP-naïve patients exhibited some form of antibiotic resistance". I recommended the author should discuss about H. pylori antibiotic resistance in a separate section.
- the author should revised the sentence "in 2017 it deemed antibiotic-resistant HP strains a serious threat to public health", WHO declared about Clarithromycin-resistance H. pylori due to efficacy of this antibiotic in determination of final cure rate. the reference number 10 was also should be changed to relevant WHO guideline.
- the information of several sections e.g. risk factor, Etiopathogenesis, as well as Clinical presentation is general. there is no interesting novel findings.
- Regarding UBT, what is difference between C13 and C14?
- the author need to discuss about advantage, limitation, as well as specificity or Sensitivity of each H. pylori diagnostic test. In addition, the application of molecular approach in detection of H. pylori as well as H. pylori antibiotic resistance is not suggested.
- the author could be used from several published meta-analysis on the efficacy of treatment regimen, CYP2C19 polymorphism, as well as immune-relevant Polymorphism on the H. pylori cure rate.
- there is insufficient data regarding application of biosensors, nanotechnology, and vaccines.
- the author should be specified conclusion with further perspective for future studies.
Round 2
Reviewer 2 Report
well revised.
Author Response
Thank you very much for your feedback!